# The Effect of Dopants on Structure Formation and Properties of Cast SHS Alloys Based on Nickel Monoaluminide

**DOI:** 10.3390/ma16093299

**Published:** 2023-04-22

**Authors:** Vitalii V. Sanin, Maksym I. Aheiev, Yury Yu. Kaplanskii, Pavel A. Loginov, Marina Ya. Bychkova, Evgeny A. Levashov

**Affiliations:** Scientific-Educational Center of SHS, National University of Science and Technology “MISIS”, Leninsky Prospect 4, Bldg. 1, Moscow 119049, Russia; sanin@misis.ru (V.V.S.); aheievmi@gmail.com (M.I.A.); kaplanskii.ii@misis.ru (Y.Y.K.); pavel.loginov.misis@list.ru (P.A.L.); bychkova@shs.misis.ru (M.Y.B.)

**Keywords:** centrifugal SHS metallurgy, heat-resistant alloys, high-temperature oxidation resistance, kinetics and mechanism of oxidation

## Abstract

Alloys based on NiAl-Cr-Co (*base*) with complex dopants (*base+2.5Mo-0.5Re-0.5Ta, base+2.5Mo-1.5Re-1.5Ta, base+2.5Mo-1.5Ta-1.5La-0.5Ru, base+2.5Mo-1.5Re-1.5Ta-0.2Ti, base+2.5Mo-1.5Re-1.5Ta-0.2Zr*) were fabricated by centrifugal SHS metallurgy. The phase and impurity compositions, structure, mechanical properties, and the mechanism of high-temperature oxidation at T = 1150 °C were studied; the kinetic oxidation curves, fitting equations and parabolic rate constant were plotted. Al_2_O_3_ and Co_2_CrO_4_ were the major phases of the oxidized layer. Three layers were formed: I—the continuous Al_2_O_3_ layer with Co_2_CrO_4_ inclusions; II—the transitional MeN-Me layer with AlN inclusions; and III—the metal layer with AlN inclusions. The positive effect of thermo-vacuum treatment (TVT) on high-temperature oxidation resistance of the alloy was observed. The total weight gain by the samples after oxidative annealing decreased threefold (from 120 ± 5 g/m^2^ to 40 ± 5 g/m^2^). The phases containing Ru and Ti microdopants, which reduced the content of dissolved nitrogen and oxygen in the intermetallic phase to the values ∑_O, N_ = 0.0145 wt.% for the *base+2.5Mo-1.5Ta-1.5La-0.5Ru* alloy and ∑_O,N_ = 0.0223 wt.% for the *base+2.5Mo-1.5Re-1.5Ta-0.2Ti* alloy, were identified by transmission electron microscopy (TEM). In addition, with the significant high-temperature oxidation resistance, the latter alloy with Ti had the optimal combination of mechanical properties (σ_ucs_ = 1644 ± 30 MPa; σ_ys_ = 1518 ± 25 MPa).

## 1. Introduction

NiAl is widely used to produce heat-resistant alloys for the components of gas-turbine engines. The drawbacks of these alloys include low mechanical strength and ductility at room temperature, resulting in insufficient manufacturability and risk of failure [1,2,3,4,5,6,7,8]. Various plasticizers are added to alloys for increasing fracture toughness [3,4,5,6,7,8,9,10]. An essential requirement on hot gas path materials is the oxidation resistance of the surface at high temperatures and under thermocyclic loading [11,12,13,14,15,16,17,18,19].

One of the known methods for producing cast and powder intermetallic-based materials is the self-propagating high-temperature synthesis (SHS) and its technological types: elemental synthesis [20,21,22,23,24,25] and centrifugal SHS casting [25,26,27,28,29]. For both of them, research aiming to optimize the composition and modes for synthesizing CompoNiAl series alloys based on NiAl-Cr-Co (*base*) is currently underway [27,28,29,30].

Microdoping with molybdenum and niobium (the *base+MoNb* alloy) obtained by elemental synthesis enhances resistance to viscoplastic deformation at temperatures above 800 °C due to the formation of the Cr_0.5_Mo_0.5_ and Cr_0.7_Mo_0.3_ phases [31,32]. High-level mechanical properties were attained at 900 °C for the studied samples after HIP: σ_ucs_ = 615 ± 9 MPa; σ_ys_ = 488 ± 7 MPa; and ε_pd_ = 62.2 ± 1.4 %.

It was found for the cast SHS alloys *base+X* (where *X = La*, *Mo*, *Ta*, *Re*, *Zr*) [32,33] Reference addedthat doping the alloy with Mo and Re resulted in the formation of a cellular eutectic structure [32]. Doping with up to 15% Mo and 1.5% Re improved mechanical properties up to the following values σ_ucs_ = 1604 ± 80 MPa, σ_ys_ = 1520 ± 80 MPa, and ε_pd_ = 0.79%; additional annealing at T = 1250 °C for 180 min enhanced the properties to σ_ucs_ = 1800 ± 80 MPa, σ_ys_ = 1670 ± 80 MPa, and ε_pd_ = 1.58%. In turn, rhenium modified the structure of the *base+15Mo1.5Re* alloy, thus enhancing mechanical properties up to the values σ_ucs_ = 1682 ± 60 MPa, σ_ys_ = 1538 ± 60 MPa, and ε_pd_ = 0.87%; additional annealing further improved them to the values σ_ucs_ = 2019 ± 60 MPa, σ_ys_ = 1622 ± 60 MPa, and ε_pd_ = 5.88%. The hierarchical three-level structure of the *base+15%Mo* alloy was identified: the first level was formed by dendritic *β*-NiAl grains with interlayers of molybdenum-containing (Ni,Co,Cr)_3_Mo_3_C and (Mo_0.8_Cr_0.2_)_x_B_y_ phases having a cell size up to 50 µm; the second level consisted of strengthening submicron-sized Cr(Mo) particles distributed along grain boundaries; and the third level comprised coherent Cr(Mo) nanoprecipitates (sized 10–40 nm) within the bulk of *β*-NiAl dendrites. Doping with interstitial elements enhanced the oxidation resistance of the β-alloy due to the formation of additional phases [33]. Volatile oxides MoO_3_, Mo_3_O_4_ and CoMoO_4_ disrupting the integrity of the protective layer were formed upon oxidation of alloys doped with molybdenum. Oxygen and nitrogen penetration depth increased with molybdenum concentration. In the tantalum-containing alloy, the Ta_2_O_5_ phase was formed in the intergrain space; this phase reduced the rate and depth of oxygen diffusion. The alloy doped with zirconium was characterized by the best high-temperature oxidation resistance; the extent of oxidation after 30 h was 21 g/m^2^. The zirconium-containing continuous upper layer Al_2_O_3_ + Zr_5_Al_3_O_0.5_ blocked the external diffusion of oxygen and nitrogen, thus increasing high-temperature oxidation resistance [33]. Aheiev et al. [33] also studied the effect of nitrogen dissolved in the alloy on the mechanism of high-temperature oxidation. Nitrogen reacts with aluminum to form aluminum nitride AlN, thus altering the oxidation mechanism. The vacuum induction melting (VIM) technique was used for degassing the alloy. The extent of oxidation after 30 h was 50 g/m^2^ for the *base* alloy and 22 g/m^2^ for the *base+VIM* alloy. However, the technological process of VIM is extensive; therefore, microdoping of alloys with metals (Ti, Zr, Hf, Re, etc.) exhibiting high chemical affinity for oxygen and nitrogen is performed to reduce the content of gas impurities. Thus, doping with titanium allowed one to bind gas impurities and noticeably reduce their content in the nickel aluminide-based solid solution, which had a positive effect on properties of the alloy [1,2]. Density increased [32] and the end products became heavier with the rising content of rhenium dopant in *β*-NiAl. Ruthenium is an expensive interstitial element but has an advantage over rhenium in terms of its density: ρ_Re_ = 21.02 g/cm^3^ and ρ_Ru_ = 12.41 g/cm^3^. Re and Ru exhibit similar effects: they stabilize the phase composition and ensure comminution of structural components, showing tendency toward low liquation during crystallization [1]. Furthermore, ruthenium is an active getter of dissolved gases. The effect of Ru on the base composition of the *base* alloy [27,28] has not been studied earlier.

Therefore, research into the effect of chemical composition on oxidation resistance, mechanical properties, and searching for the optimal compositions of the *β*-NiAl-based alloy is rather relevant today. This study aimed to produce cast NiAl-Cr-Co (*base*) alloys doped with complex additives by centrifugal SHS metallurgy and investigate the features of their microstructure, mechanical properties, kinetics and the mechanism of oxidation.

## 2. Materials and Methods

The calculated compositions of the alloys in the *base+X* system are listed in Table 1.

Synthesis was carried out using a radial-type centrifugal setup under high gravity conditions (up to 150 g). The general scheme of the employed centrifugal setup was provided in refs. [23,24,25,26].

The design of the setup allows one to specify the number of revolutions of the centrifuge rotor in a controlled manner to ensure the desired acceleration level. A distinctive feature of this technology is that the relatively available oxide feedstock is used and high combustion temperature (2100–3500 °C) is attained, so the phase of the target product can be separated from the cinder phase. The chemical scheme of the process can be represented as:NiO + Cr_2_O_3_ + Co_3_O_4_ + MoO_3_ + Al + (X) + (FA) –> [NiAl-Cr-Co-Mo-(X)] + Al_2_O_3_(1)
where: FA (functional additive) is CaF_2_, Na_3_[AlF_6_], etc; X is Me (Zr, Ta, Re, Ru, Ti, La).

Table 2 lists grades and properties of the initial powders. Dopants were added to the reaction mixture so as to obtain the desired composition of the alloy.

The scheme for preparing exothermic mixtures included drying the components in SNOL-type drying ovens at 90 °C during 1 h, dosing reagents, mixing and casting the mixture in graphite molds. Mixing was carried out in an MP4/0.5 planetary mill with a 1 L drum during 15–20 min at the ball-to-powder ratio of 1:5. The combustion temperature of the mixtures was higher than the melting point of the end products of synthesis, enabling complete phase segregation due to gravity separation of the metallic melt and cinder (Figure 1). The Zr, Ta, Re, Ru and La components were added to the reaction mixture as pure elements, while Mo was added as MoO_3_ oxide. Aluminum was used to reduce the oxide charge. Different grades of aluminum are used to control SHS processes [25].

Thermodynamic calculations of the adiabatic combustion temperature (T_f_) were preliminarily performed using the THERMO software ver.3 package. T_f_ was 2300–2400 °C for all the compositions under study, being noticeably higher than the melting points of the synthesis products. The optimal acceleration values for these systems were identified earlier and were equal to 140–150 g [32]. The highest yield of the end product into a metal ingot was attained at this acceleration value. At least three ingots 80 mm in diameter and 25–30 mm high were synthesized for each composition (Table 1), making it possible to study homogeneity and reproducibility of the results. According to the recommendations [32], thermo-vacuum treatment (TVT) of the ingots was carried out at 1250 °C for 2 h to enhance ductility and residual strain. TVT led to strain relaxation, and the recrystallization of grains of the major *β*-NiAl phase and excess phases. Furthermore, TVT contributed to the degassing of dissolved or adsorbed impurities (nitrogen and oxygen) [33].

A Thermo Fisher Scientific Finnigan Element glow discharge mass spectrometer and a double-focusing spectrum analyzer (in the Nier–Johnson geometry) were employed for quantitative analysis of major components and impurities. This instrument was used to identify the chemical composition of both the base and impurity elements.

The phase composition was determined by X-ray phase analysis (XRD) on a D2 PHASER diffractometer (Bruker AXS GmbH, Mannheim, Germany) using Cu-Kα radiation in the 2*θ* range = 10–140°.

The microstructural studies were carried out on an S-3400N scanning electron microscope (Hitachi, Tokyo, Japan) coupled to a NORAN System 7 X-ray microanalysis system (Thermo Scientific, Waltham, MA, USA), as well as a JEM-2100 transmission electron microscope (TEM) (Jeol, Tokyo, Japan) using a Gatan 650 Single Tilt Rotation Analytical Specimen Holder (Gatan, Inc., Pleasanton, CA, USA). The samples (lamellae) were fabricated from the preliminarily prepared foil using the focused ion beam technique on a Quanta 200 3D FIB instrument (FEI Company, Hillsboro, OR, USA). TEM foils were prepared by ion etching on PIPS II system (Gatan, Inc., Pleasanton, CA, USA).

Compression tests were carried out on an LF-100KN universal testing machine (Walter+Bai AG, Löhningen, Switzerland) at room temperature in compliance with the State Standard GOST 25.503-97.

Oxidative annealing was carried out in air at 1150 °C during 30 h in an SShOL 1.1.6/12-M3 laboratory pit-type electric furnace; the samples were weighed periodically. Changes in sample weight normalized to the unit surface area over a certain time period were determined. The experimental data were used to plot the oxidation curves and fitting equations. Samples 8 mm in diameter and 4 mm high were cut on an EDM machine GX-320L (CHMER EDM, Taichung, Taiwan) and ground to a roughness of Rz = 5.

## 3. Results and Discussion

The low ductility of intermetallic alloys at room temperature impedes their practical application for fabricating geometrically complex items. The presence of detrimental impurities is an additional factor deteriorating properties of the alloys. Therefore, an important problem is controlling the chemical composition and impurity content in alloys. Chemical analysis of the synthesized ingots showed that they corresponded to the calculated compositions for the major components of the alloy. Table 3 lists the impurity composition of *base-X* alloys. The total contents of Mg, Na, Si, Ca, K, Fe Mn, Cu, W, S and C impurities are provided.

Impurity elements Mg, Na, Si, Ca, K, Mn and Cu are the accompanying ones and are transferred to the synthesis products from the starting reagents. The total impurity content is 0.15 ± 0.02 wt.%, which lies within the acceptance region for heat-resistant nickel alloys. Meanwhile, technical solutions that would reduce their concentration need to be found. The excessive content of carbon (up to 0.017 wt.%) in all the samples is due to the fact that SHS was performed in graphite molds. Most of La participates in the reaction of oxide reduction, being regarded as a competitor of the major reducing agent (Al). Therefore, La content in the ingot of alloy 3 (*base+2.5Mo-1.5Ta-1.5La-0.5Ru)* was decreased (0.86 wt.%) compared to the calculated value. Importantly, the contents of oxygen and nitrogen impurities decline to the value ∑_O,N_ = 0.0145 wt.% for the *base+2.5Mo-1.5Ta-1.5La-0.5Ru* alloy and ∑_O,N_ = 0.0223 wt.% for the *base+2.5Mo-1.5Re-1.5Ta-0.2Ti* alloy. Ruthenium and titanium act as getters of oxygen and nitrogen, thus exhibiting a positive effect on strength properties and high-temperature oxidation resistance of the alloy.

Figure 2 and Table 4 show the phase composition of the synthesized alloys. β-NiAl was the major phase. The Ni(Al,Ta) phase was present in alloys 2 and 3 with Ta content up to 1.5%, while alloy 3 contained the MoNi phase. The Ni(Al,Ti) phase was identified in alloy 4 doped with 0.2% Ti, while doping with 0.2% Zr (alloy 5) gave rise to the Ni_2_(Zr,Al) phase.

Figure 3 shows that in the case of complex doping of an alloy with Mo, Re and Ta metals, inclusions based on solid solution of chromium of different compositions were formed in the *β*-NiAl matrix. For the *base+2.5Mo-0.5Re-0.5Ta* composition, globular micron- and submicron-sized inclusions based on solid solution of chromium reside inside *β*-NiAl grains. Inclusions 2–8 µm thick with compositions (Cr)_Ni,Al,Mo,Re_, (Cr)_Mo,Ta_ and (Cr)_Mo_ were formed in the intergrain space.

As Ta and Re concentrations were increased to 1.5%, the (*base+2.5%Mo-1.5%Re-1.5%Ta*) alloy acquired a well-defined mesh structure (Figure 4). Cr- and Re-based solid solutions were contained within intergrain interlayers. The Ni(Al,Ta) phase was located at grain boundaries between the solid solution and the NiAl matrix. Submicron-sized NiAl precipitates were detected inside the interdendritic layers. As demonstrated by Aheiev et al. [31], tantalum localization along the boundaries of the major phase grains had a positive effect on strength properties and increased plastic strain at room temperature. Stringed precipitates of α-Cr were also revealed in the NiAl matrix.

In the *base+2.5%Mo-1.5%Ta-1.5%La-0.5%Ru* alloy (Figure 5), a multicomponent eutectic system was formed between dendrite branches. MoNi precipitates resided in the center of the intergrain space. The (Cr)_Ni,Co,Mo_ solid solution was formed around the MoNi phase. Furthermore, the intergrain space hosted a chromium-based solid solution with dissolved Mo, Ta, Ru and La. Identically to the alloy described previously, the Ni(Al,Ta) intermetallic phase was formed at grain boundaries, and stringed inclusions of α-Cr were observed inside the grain.

Additional doping of the alloy with titanium (*base+2.5%Mo-1.5%Re-1.5%Ta-0.2%Ti*) did not qualitatively alter the structure (Figure 6). The Ni(Al,Ti) was formed at grain boundaries, which increased plastic strain resistance and strength of the alloy as it can be seen in Table 5. The interdendritic layer hosted the chromium-based solid solution, while the matrix contained stringed inclusions of α-Cr.

In a similar manner, doping the alloy with 0.2% Zr (*base+2.5%Mo-1.5%Re-1.5%Ta-0.2%Zr*) resulted in the precipitation of the Ni_2_(Zr,Al) phase (Figure 7). The intergrain layers were formed by solid solution based on chromium and rhenium with NiAl precipitates. Microprobe analysis showed that tantalum was a component of the (Re)_Cr,Mo,Ta_ solid solution.

Table 5 lists the results of measuring the mechanical properties of cast SHS alloys. One can see that the *base+2.5%Mo-1.5%Re-1.5%Ta-0.2%Ti* alloy had the best combination of properties (hardness, strength, the yield point and residual strain): σ_ucs_ = 1644 ± 30 MPa, σ_ys_ = 1518 ± 25 MPa (Figure 8).

In order to study the effect of complex doping on the high-temperature oxidation resistance of the alloys, air annealing at 1150 °C was performed during 30 h; the samples were periodically weighed. When investigating the oxidative resistance for the previous series of alloys, Aheiev et al. [31] demonstrated that a multilayer oxide film was formed in the alloys characterized by the greatest weight gain. The transitional MeN-Me layer mainly consisted of nitrides due to the high content of dissolved nitrogen, which diffused from the alloy along grain boundaries to interact with aluminum, thus disrupting alloy integrity. Therefore, additional vacuum annealing (TVT at 700 °C during 2 h) was carried out in this study for the *base+2.5Mo-1.5%Re-1.5%Ta* alloy as an example.

Table 6 lists the weight gain values after oxidative annealing at 1150 °C during 30 h, as well as the kinetic regression equations corresponding to the oxidation curves shown in Figure 9. Figure 9b,d shows the parabolic rate constant determined for each composition. The parabolic rate constant *k_p_* was measured by the method as follows (2). Line 2.2 corresponds to the sample subjected to additional vacuum annealing. The parabolic rate constant of oxidation is shown in Figure 9b,d. The parabolic rate constant was measured as follows:(2)$${\left(\frac{\Delta m}{s}\right)}^{2}=k_{p}t$$
where: ∆*m* is the mass change; *S* is the surface area; *t* is time.

For samples 1, 2.2, 3, 4 and 5, the shape of the curves corresponds to the parabolic law of oxidation: a continuous barrier oxide layer was formed within the first few hours of oxidation, inhibiting oxygen diffusion into the sample. The oxidation curve of sample 2.1 is fitted by logarithmic law (local phase segregation takes place due to the internal stress emerging during thermal cycling).

Figure 10 shows the appearance of the oxidized samples. Each alloy had its own hue and certain topology. Signs of physical degradation were observed for none of the samples.

Table 7 summarizes the results of XRD analysis of the oxidized layer of the samples; the XRD spectra are shown in Figure 11. Al_2_O_3_ was the main oxidation product. Co_2_CrO_4_ was the next phase (according to its weight fraction). Lines belonging to NiAl were also present in the XRD spectra along with those corresponding to oxides. Traces of the Ni_3_AlN phase were detected among the oxidation products with composition 1.

Figure 12 shows the microstructures of the samples after high-temperature oxidation resistance tests, the size of oxide and transitional layers being specified. One can see that in all the cases, a continuous oxide film impeding diffusion-controlled penetration of oxygen and nitrogen to the alloy was formed at the initial oxidation stage. A transitional MeN-Me layer was formed in the alloys with increased Re content (1.5%), being indicative of nitrogen diffusion in the alloy.

A thorough analysis of the oxidation mechanism is provided in Figure 13, Figure 14, Figure 15, Figure 16, Figure 17, Figure 18, Figure 19, Figure 20, Figure 21, Figure 22 and Figure 23. Alloy 2.2 containing Mo, Re and Ta microdopants was chosen as an example of the positive effect of thermo-vacuum treatment of cast SHS alloys on the microstructure after high-temperature oxidation resistance testing; its analysis is provided in Figure 15.

The top dense oxidized layer on the *base+2.5%Mo-0.5%Re-0.5%Ta* sample with composition 1 consisted of Al_2_O_3_, which was a distinctive feature of this alloy compared to other ones. The Al_2_O_3_ alloy was formed during the initial oxidation period and impeded oxygen diffusion into the sample. The MoNi phase and the chromium-based solid solution resided in the oxidized layer as globular inclusions and around non-oxidized NiAl regions (Figure 13). Because of the small dopant concentration in the alloy with composition 1, the sample had a structure with a low content of solid-solution inclusions in the intergrain space. This factor could restrain oxygen and nitrogen diffusion at grain boundaries during the initial oxidation period, contributing to the formation of solid Al_2_O_3_ film, as well as preventing nitrogen diffusion and the formation of aluminum nitride. Meanwhile, nitrogen dissolved in the alloy entered the reaction yielding Ni_3_AlN nitride perovskite.

After oxidative annealing, the *base+2.5%Mo-1.5%Re-1.5%Ta* alloy with composition 2 (characterized by increased Re and Ta contents) had a multilayer surface structure. According to the pattern of oxygen and nitrogen distribution, the oxide layer could be subdivided into three sublayers: *I*—the continuous Al_2_O_3_ oxide film with a non-uniform distribution of Co_2_CrO_4_ spinel inclusions; *II*—the transitional MeN-Me layer containing AlN inclusions; and *III*—the metal layer with sparse inclusions of AlN (Figure 14). The oxide layer was formed at the initial stage during the formation of chromium, cobalt and aluminum oxides and Co_2_CrO_4_ spinel. The Co_2_CrO_4_ phase resulted from interaction between chromium and cobalt oxides. Spinel was composed of coarse grains with defects through which surface diffusion of oxygen and nitrogen into the sample could occur. In turn, the formation of Al_2_O_3_ reduced the partial pressure of oxygen and impeded spinel formation [34]. A continuous Al_2_O_3_ layer was located at the boundary between sublayers *I* and *II*, being responsible for the enhancement of oxidation resistance. Sublayer *I* contained, along with oxides, a small amount of the Ni_3_Al phase that had been formed during the oxidation of aluminum. In the transitional MeN-Me layer, nitrogen reacted with aluminum to yield AlN nitrides, which also depleted NiAl to Ni_3_Al. Along with Ni_3_Al, the interaction between nitrogen and aluminum contributed to the formation of the MoNi intermetallic phase [34].

In order to assess the effect of thermo-vacuum treatment on high-temperature oxidation resistance, we analyzed the microstructure of the oxidized alloy with composition 2 that had undergone vacuum annealing (Figure 15). Figure 12 (No. 2.2: *base+2.5%Mo-1.5%Re-1.5%Ta+TVT*) showed that the maximum thickness of the alloy after vacuum annealing was lower (135 µm) than that for the sample not subjected to TVT (160 µm). The average thickness of the oxidized layer for the sample after TVT was 50 µm. Weight gain was also approximately threefold lower (Table 6), thus proving the positive effect of vacuum annealing. Not only did nitride formation during the initial oxidation stage disrupt the integrity of ingots, but it also impeded the formation of the Al_2_O_3_ barrier oxide layer [35,36]. The lower content of nitrogen impurity inhibited the formation of nitrides in the alloy, thus increasing its oxidation resistance. An analysis of the alloy microstructure after TVT showed that the structure and the mechanism of formation of the oxidized layer were similar to those for the alloy not subjected to TVT: three sublayers were formed, NiAl became Al-depleted, and the Ni_3_Al and MoNi phases were precipitated [36].

Figure 16 shows the structure of the oxidized surface of the sample with composition 3 (*base+2.5%Mo-1.5%Ta-1.5%La-0.5%Ru*). The content of Co_2_CrO_4_ spinel in this sample was significantly lower (13.4 wt.%) (Table 6). The oxidized layer contained Al_2_O_3_ with chaotic inclusions of the Co_2_CrO_4_ phase. A distinctive feature of this alloy is that the Co_2_CrO_4_ phase was located under the Al_2_O_3_ layer [37,38]. Globular inclusions of chromium and unreacted regions of the TaCo_2_ phase and (Cr)_Ni,Mo_ solid solution were detected in the oxidized layer. This sample did not contain the Ni_3_Al phase; AlN inclusions resided at the NiAl/(Cr)_Ni,Mo_ interface.

A detailed analysis of the oxidized layer of the *base+2.5%Mo-1.5%Ta-1.5%La-0.5%Ru* sample was carried out by transmission electron microscopy (TEM). A lamella cut from a cross-section of the metal/oxidized layer interface by FIB was used as a study object (the lamella cutting site is indicated in Figure 16). Figure 17, Figure 18 and Figure 19 and Table 8 show the images of the lamella and the EDS maps of distribution of the respective elements. The near-surface layer consisted of Al_2_O_3_ (Figure 17, spectra 2 and 6), which was an efficient barrier blocking oxygen and nitrogen diffusion deep inside the sample. Nanosized inclusions of Co_2_CrO_4_ spinel with fcc crystal lattice (space group Fd3m) and lattice parameter *a* = 8.131 Å were arranged along the boundaries of Al_2_O_3_ grains (Figure 18). As found earlier, this phase present as coarse grains negatively affected the high-temperature oxidation resistance of alloys, since they caused the formation of cracks acting as oxygen diffusion channels. However, in the *base+2.5%Mo-1.5%Ta-1.5%La-0.5%Ru* alloy, the Co_2_CrO_4_ phase consisted of uniformly distributed particles sized up to 100 nm and did not cause cracking of the oxide layer. The electron diffraction data recorded for the surface region of the oxidized layer proved that nanosized Co_2_CrO_4_ spinel crystals were present (Figure 18). Under the layer based on Al_2_O_3_ and Co_2_CrO_4_, there was an interlayer consisting of coarse grains of chromium-based solid solution. It was shown by EDS that Co, Ni and Mo (15–20 wt.%) were dissolved in this phase. The substrate region immediately adjacent to the oxidized layer consisted of the Ni_3_Al phase formed via Ni depletion in NiAl (Figure 17 and Figure 18, spectrum 8). Lamellar AlN inclusions sized <1 µm (Figure 17 and Figure 18, spectra 3 and 4) were detected at the Ni_3_Al/MoNi interface (Figure 17 and Figure 18, spectra 3 and 4). They were formed via the diffusion of nitrogen dissolved in the alloy along grain boundaries towards the sample surface and interaction with aluminum.

Ru is an efficient getter of oxygen and nitrogen [2]. Doping the alloy with this element significantly reduced the weight gain by the samples in oxidative testing. It is most likely that as oxygen diffused deep into the sample, Ru dissolved in the (Cr) phase formed stable complex oxide. The examination of the lamella revealed inclusions of this oxide. The small size of inclusions did not allow us to accurately identify the phase according to the electron diffraction data (Figure 18). However, relying on the chemical composition of this region, a hypothesis can be put forward that this oxide was chromium–ruthenium double oxide (Cr_x_Ru_y_)O_2_.

Similar to alloy 2, the oxidized surface of the alloy with composition 4 (*base+2.5%Mo-1.5%Re-1.5%Ta-0.2%Ti*) (Figure 20) consisted of three layers. The top 40-µm-thick oxide layer composed of Al_2_O_3_ and Co_2_CrO_4_ spinel was characterized by low density and high pore content. Below, there was a thin continuous sublayer (5 µm) composed of Al_2_O_3_, which impeded oxygen penetration inside the material. A thick layer (up to 100 µm) based on AlN with inclusions of chromium-containing phases (Cr,Co)Ni, (Cr)_MoRe_ and (Cr)_Mo_ lay at the boundary with the substrate.

A lamella cut from the transverse section of the transitional MeN-Me layer in which nitrides are formed was studied by TEM (the lamella cutting site is shown in Figure 20). The structure of the lamella is presented in Figure 21 and Figure 22 and Table 9. Figure 21 also shows the EDS element distribution maps, where Ti inclusions were detected (spectra 1, 2 and 9). Aluminum nitride AlN, having the hexagonal crystal lattice (space group P6_3_mc) and lattice parameters *a* = 3.083 Å, *c* = 5.001 Å, was the major phase of the transitional layer (Figure 21, spectra 3, 7 and 8). Nitrogen diffusing along boundaries of the grains of the loose Al_2_O_3_ + Co_2_CrO_4_ oxide layer into the metal, as well as nitrogen impurity dissolved in the alloy, reacted with aluminum contained in the matrix to give rise to AlN. Local Al depletion in the alloy yielded chromium (Cr)-based solid solution containing Ti, Co, Ni, Mo and Re at concentrations ranging from 7 to 25 wt.% (Figure 21, spectra 1, 2, 4, 5 and 6).

Submicron-sized inclusions of the fcc phase TiN with lattice parameter *a* = 4.205 Å were detected at the boundaries of the grains of chromium (Cr)-based solid solution and the Cr/aluminum nitride (AlN) interface (Figure 22). Not only did the resulting TiN bind nitrogen dissolve in the alloy, thus reducing its concentration (Table 3), but it also enhanced the activity of aluminum diffusing towards the surface, which contributed to the formation of a dense oxide layer [33,39].

The oxidized layer on the surface of the *base+2.5%Mo-1.5%Re-1.5%Ta-0.2%Zr* alloy had a structure identical to those of alloys 2 and 4. Its average thickness was 95 µm. A loose layer composed of a mixture of the Al_2_O_3_ + Co_2_CrO_4_ phases and a dense interlayer consisting of Al_2_O_3_ were formed on the surface (Figure 23). The significant high-temperature oxidation resistance was presumably ensured by nanosized inclusions of the Zr_5_Al_3_O_0.5_ phase as demonstrated earlier [32] for nickel monoaluminide doped with 0.5% Zr. The MoNi phase and (Cr)_Re,Zr_, (Cr)_Re_ and (Cr)_Co,Ni_ solid solutions were observed in the transitional layer. The metal layer of the NiAl matrix also contained AlN inclusions, being indicative of nitrogen diffusion along grain boundaries.

## 4. Conclusions

Cast alloys based on NiAl-Cr-Co (*base+*) with complex dopants added (*base+2.5Mo-0.5Re-0.5Ta, base+2.5Mo-1.5Re-1.5Ta, base+2.5Mo-1.5Ta-1.5La-0.5Ru, base+2.5Mo-1.5Re-1.5Ta-0.2Ti*, and *base+2.5Mo-1.5Re-1.5Ta-0.2Zr*) were fabricated by centrifugal SHS metallurgy.The chemical composition was found to be consistent with the calculated one. The total content of impurity elements was 0.15 ± 0.02 wt.% and lay within the acceptance region. Due to binding of dissolved oxygen and nitrogen to form oxides and nitrides, doping with Ti and Ru reduced the negative role of gas impurities and enhanced high-temperature oxidation resistance of the alloy.The kinetics and the mechanism of oxidation of alloys at T = 1150 °C were studied; the kinetic regression equations describing the oxidation law were plotted. Al_2_O_3_ and Co_2_CrO_4_ are the major phases in the oxidized layer. Three layers were found to be formed: *I*—the continuous Al_2_O_3_ layer with Co_2_CrO_4_ inclusions; *II*—the transitional MeN-Me layer with AlN inclusions; and *III*—the metallic layer with local AlN inclusions.The positive effect of vacuum pre-annealing of ingots on their high-temperature oxidation resistance was observed for the *base+2.5Mo-1.5Re-1.5Ta+(TVT)* alloy as an example. The total weight gain of the annealed samples after the tests decreased threefold: from 120 ± 5 g/m^2^ to 40 ± 5 g/m^2^.Phases containing ruthenium and titanium, which reduce the content of gas impurities in the *base+ 2.5Mo-1.5Ta-1.5La-0.5Ru* alloy to the value ∑_O,N_ = 0.0145 wt.% and the *base+2.5Mo-1.5Re-1.5Ta-0.2Ti* alloy to the value ∑_O,N_ = 0.0223 wt.%, were identified by TEM.The *NiAl-12Cr-6Co-2.5Mo-1.5Re-1.5Ta-0.2Ti* alloy was found to have the optimal composition in this experimental series; it was characterized by strength properties σ_ucs_ = 1644 ± 30 MPa, σ_ys_ = 1518 ± 25 MPa and the total weight gain after oxidation of 52 g/m^2^.

## Figures and Tables

**Figure 1 materials-16-03299-f001:**
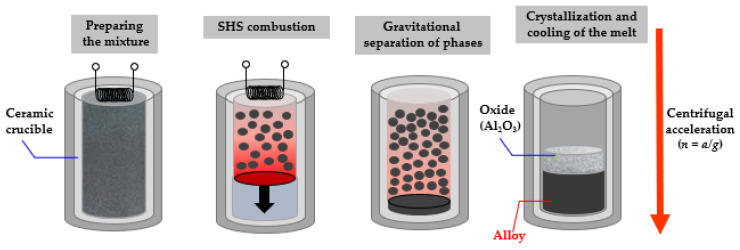
Scheme of the SHS process.

**Figure 2 materials-16-03299-f002:**
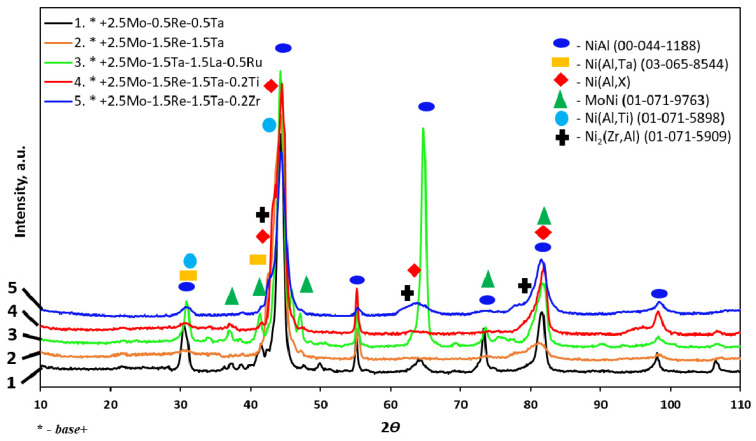
X-ray diffraction patterns of the *base+* alloys doped with *Re*, *Ta*, *La*, *Ru*, *Ti* and *Zr*.

**Figure 3 materials-16-03299-f003:**
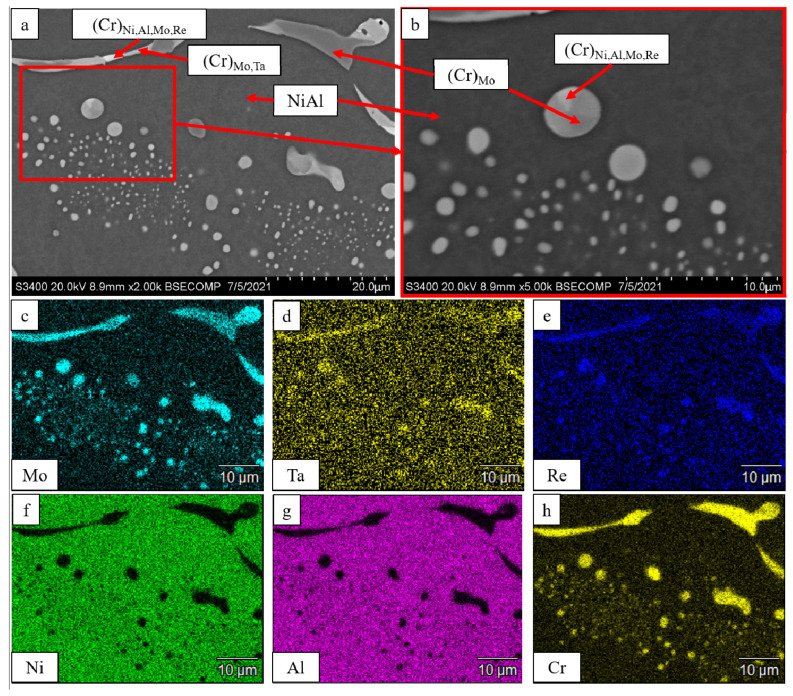
A SEM image of the *base+2.5Mo-0.5Re-0.5Ta* alloy under magnification: (**a**) ×2000; (**b**) ×5000; and the distribution maps of the major doping elements: (**c**) Mo; (**d**) Ta; (**e**) Re; (**f**) Ni; (**g**) Al, (**h**) Cr.

**Figure 4 materials-16-03299-f004:**
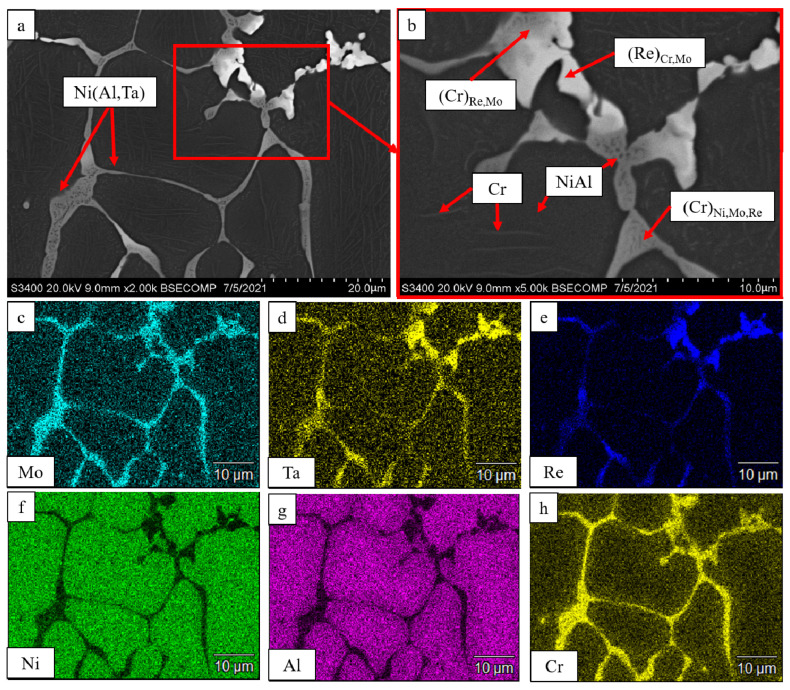
A SEM image of the *base+2*.*5Mo-1*.*5Re-1*.*5Ta* alloy under magnification: (**a**) ×2000; (**b**) ×5000; and the distribution maps of the major doping elements: (**c**) Mo; (**d**) Ta; (**e**) Re; (**f**) Ni; (**g**) Al, (**h**) Cr.

**Figure 5 materials-16-03299-f005:**
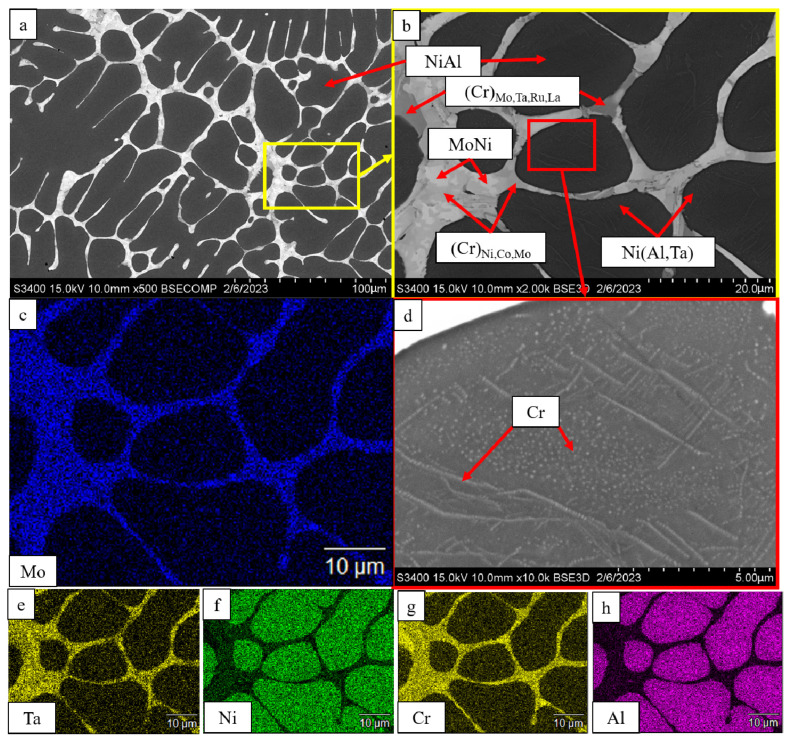
A SEM image of the *base+2.5Mo-1.5Ta-1.5-La-0.5Ru* alloy under magnification: (**a**) ×500; (**b**) ×2000; (**d**) ×10,000; and the distribution maps of the major doping elements: (**c**) Mo; (**e**) Ta; (**f**) Ni; (**g**) Cr; (**h**) Al.

**Figure 6 materials-16-03299-f006:**
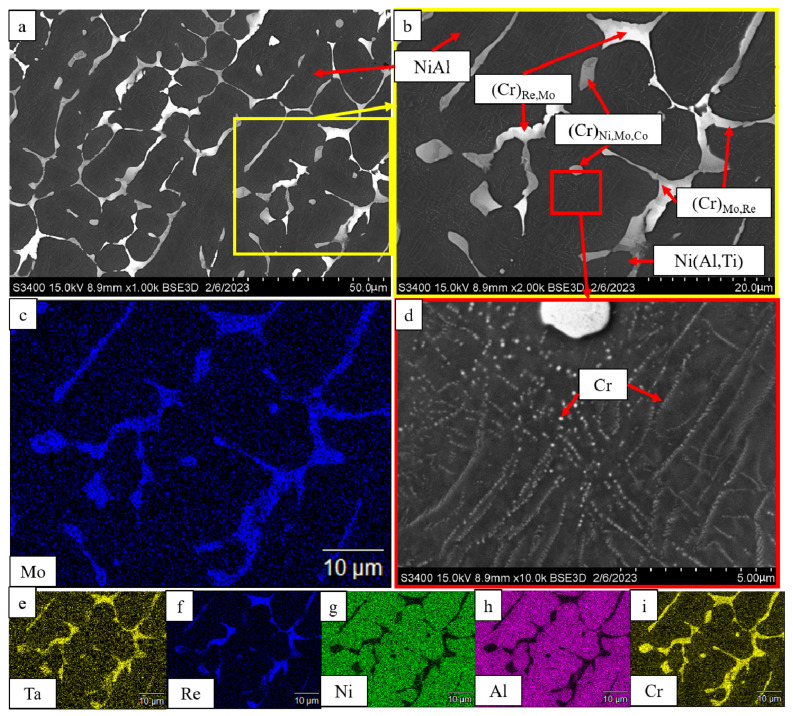
A SEM image of the *base+2.5Mo-1.5Re-1.5Ta-0.2Ti* alloy under magnification: (**a**) ×1000; (**b**) ×2000; (**d**) ×10,000, and the distribution maps of the major doping elements: (**c**) Mo; (**e**) Ta; (**f**) Re; (**g**) Ni; (**h**) Al, (**i**) Cr.

**Figure 7 materials-16-03299-f007:**
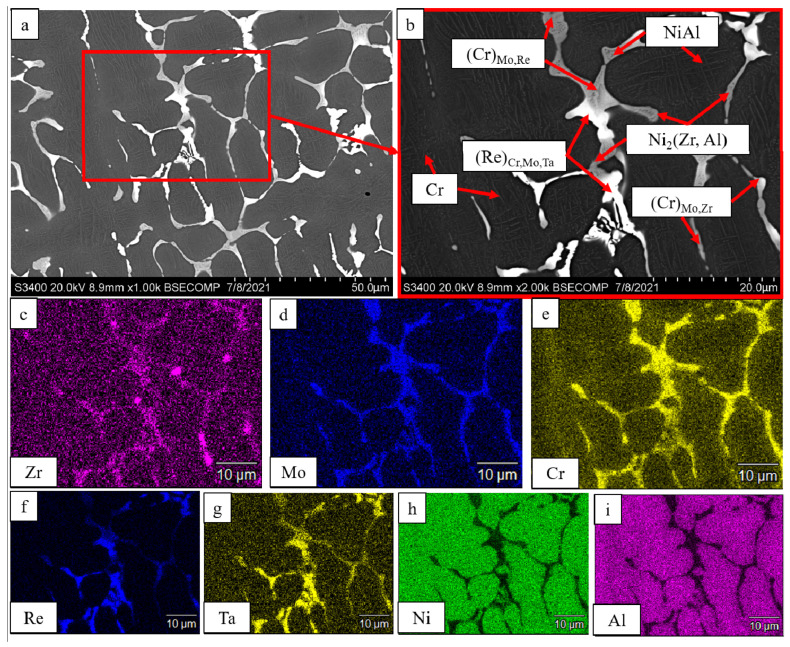
A SEM image of the *base+2.5Mo-1.5Re-1.5Ta-0.2Zr* alloy under magnification: (**a**) ×1000; (**b**) ×2000; and the distribution maps of the major doping elements: (**c**) Zr; (**d**) Mo; (**e**) Cr; (**f**) Re; (**g**) Ta; (**h**) Ni; (**i**) Al.

**Figure 8 materials-16-03299-f008:**
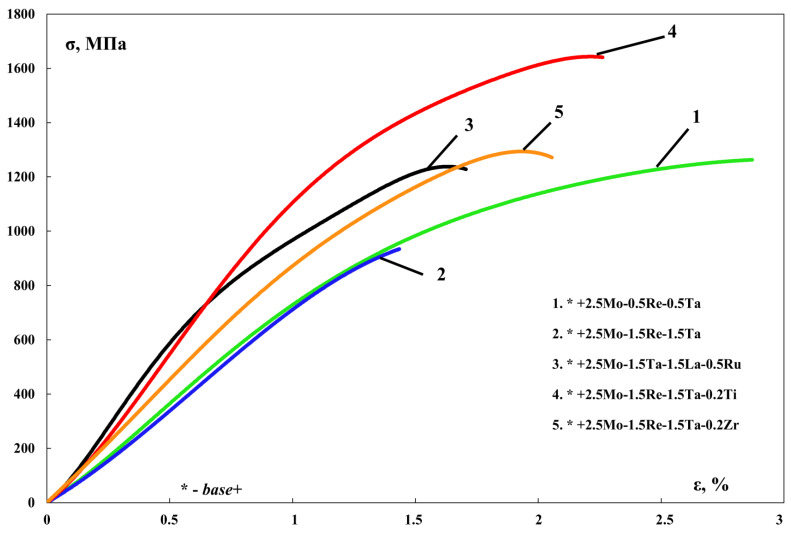
Compressive stress–strain curves for the *base-X* alloys with compositions 1–5.

**Figure 9 materials-16-03299-f009:**
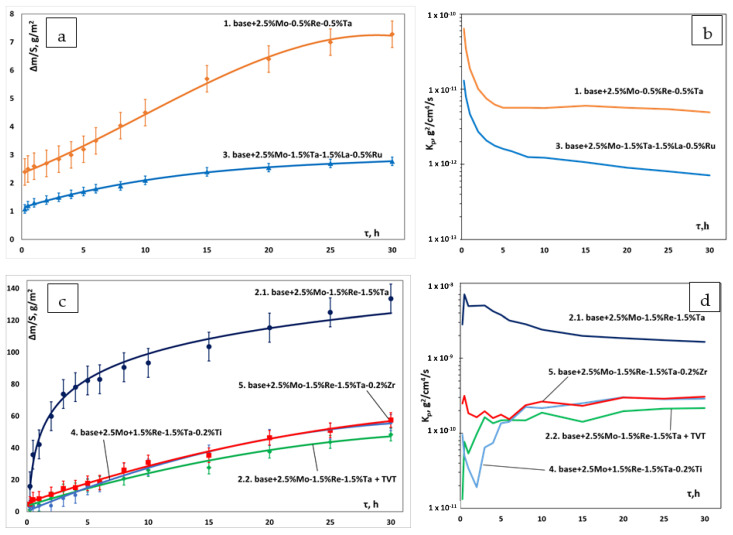
Kinetic curves of oxidation at 1150 °C for the samples of *base-X* alloys with compositions 1–5: (**a**,**c**) the weight gain as a function of oxidation duration; and (**b**,**d**) the parabolic rate constant.

**Figure 10 materials-16-03299-f010:**
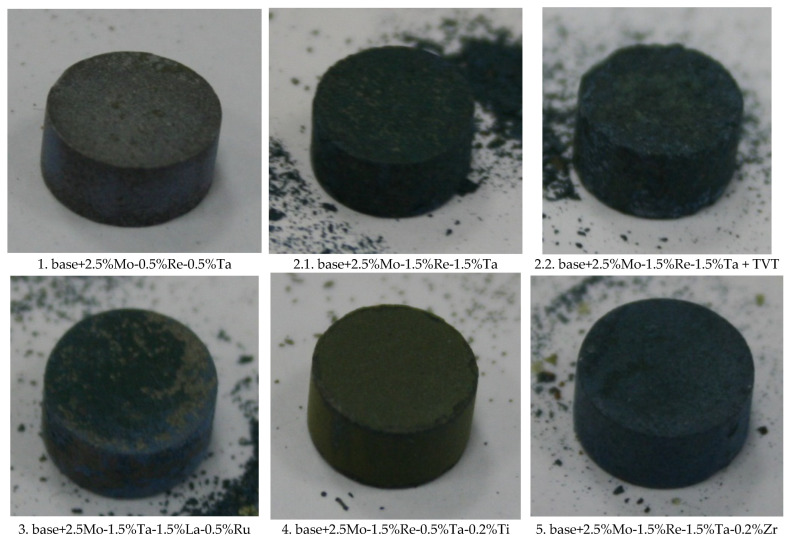
Appearance of the *base+X* samples with compositions 1–5 exposed to high-temperature oxidation resistance testing at 1150 °C during 30 h.

**Figure 11 materials-16-03299-f011:**
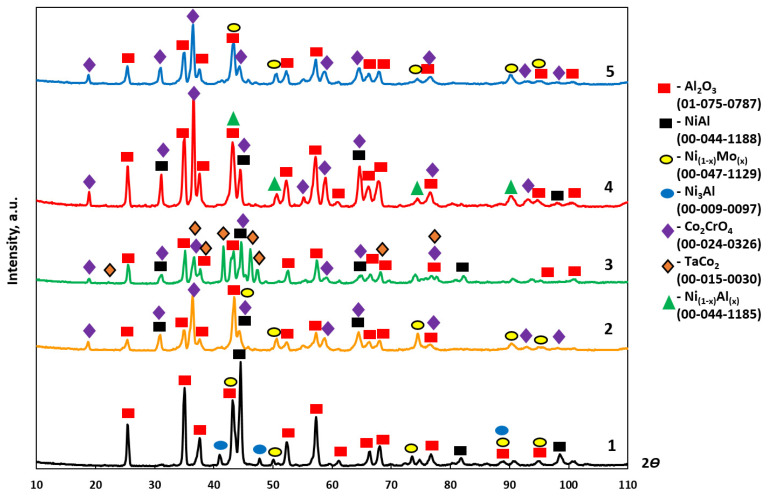
XRD spectra of the oxidized surface of the *base+X* samples with compositions 1–5 after high-temperature oxidation resistance testing at 1150 °C during 30 h: (1) X = 2.5%Mo-0.5%Re-0.5%Ta; (2) X = 2.5%Mo-1.5%Re-1.5%Ta; (3) X = 2.5%Mo-1.5%Ta-1.5%La-0.5%Ru; (4) X = 2.5%Mo-1.5%Re-1.5%Ta-0.2%Ti; and (5) X = 2.5%Mo-1.5%Re-1.5%Ta-0.2%Zr.

**Figure 12 materials-16-03299-f012:**
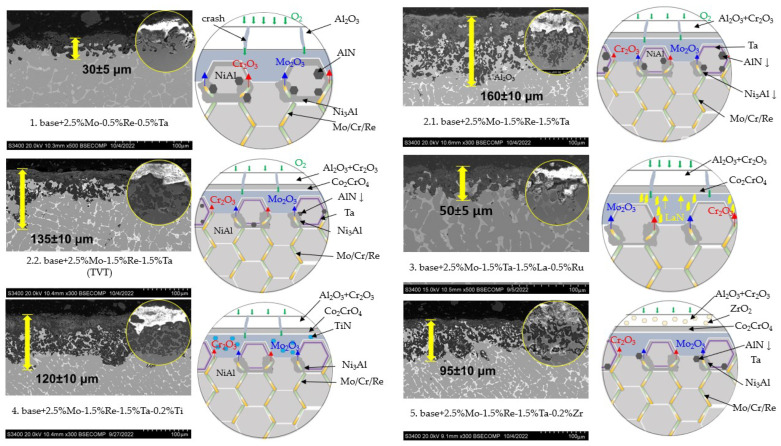
A SEM image and oxidization scheme of the *base-X* samples with compositions 1–5 after high-temperature oxidation resistance testing at 1150 °C during 30 h. An arrow shows the visible depth of the oxidized layer.

**Figure 13 materials-16-03299-f013:**
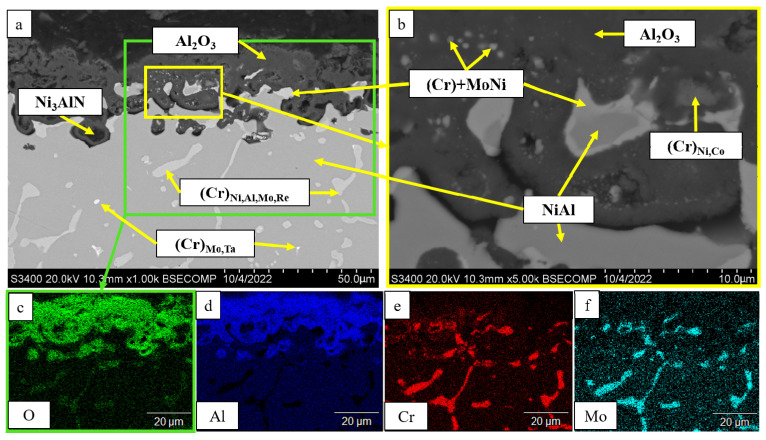
A SEM image of the surface of the oxidized *base+2.5%Mo-0.5%Re-0.5%Ta* sample with composition 1 under magnification: (**a**) ×1000; (**b**) ×5000; and the element distribution maps: (**c**) O; (**d**) Al; (**e**) Cr; (**f**) Mo.

**Figure 14 materials-16-03299-f014:**
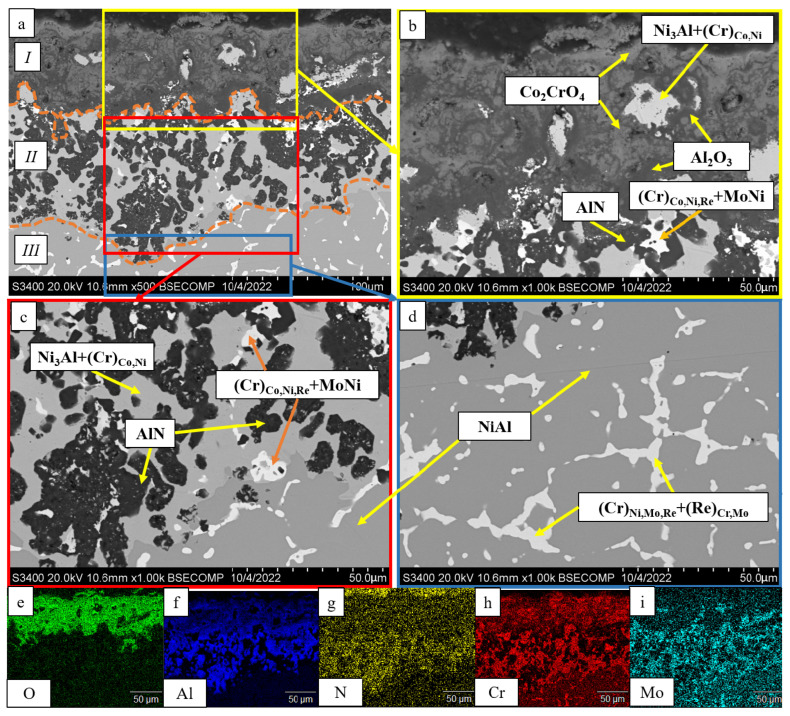
A SEM image of the surface of the oxidized sample with composition 2 (*base+2.5%Mo-1.5%Re-1.5%Ta*): (**a**) under magnification ×500; (**b**) the continuous Al_2_O_3_ oxide film with non-uniform distribution of Co_2_CrO_4_ spinel inclusions; (**c**) the transitional MeN-Me layer containing AlN inclusions; (**d**) the metal layer; and the element distribution maps: (**e**) O; (**f**) Al; (**g**) N; (**h**) Cr; (**i**) Mo.

**Figure 15 materials-16-03299-f015:**
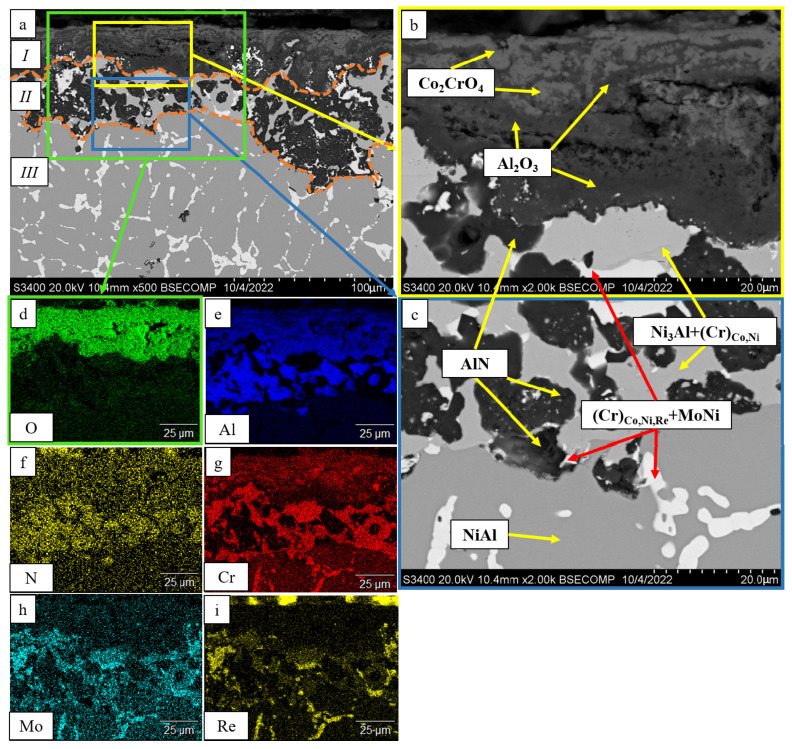
A SEM image of the surface of the oxidized sample with composition 2 (*base+2.5%Mo-1.5%Re-1.5%Ta*) after vacuum annealing: (**a**) under magnification ×500; (**b**) the continuous Al_2_O_3_ oxide film with non-uniform distribution of Co_2_CrO_4_ spinel inclusions; (**c**) the transitional MeN-Me layer containing AlN inclusions; and the element distribution maps: (**d**) O; (**e**) Al; (**f**) N; (**g**) Cr; (**h**) Mo; (**i**) Re.

**Figure 16 materials-16-03299-f016:**
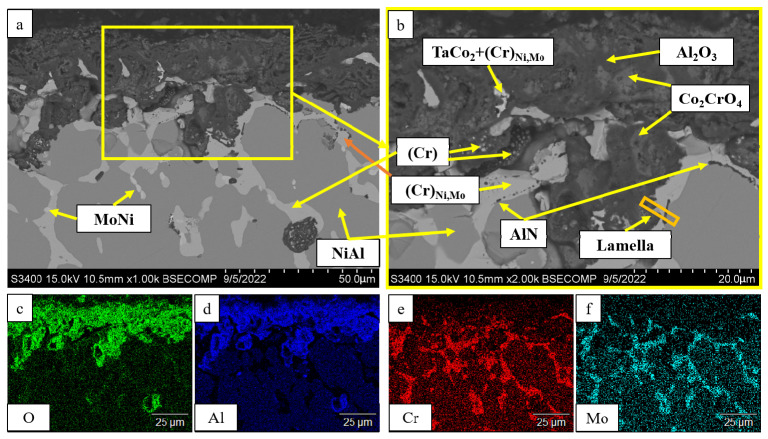
A SEM image of the surface of the oxidized sample with composition 3 (*base+2.5%Mo-1.5%Ta-1.5%La-0.5%Ru*) under magnification: (**a**) ×1000; (**b**) ×2000; the element distribution maps: (**c**) O; (**d**) Al; (**e**) Cr; (**f**) Mo; and (**b**) the lamella cutting site.

**Figure 17 materials-16-03299-f017:**
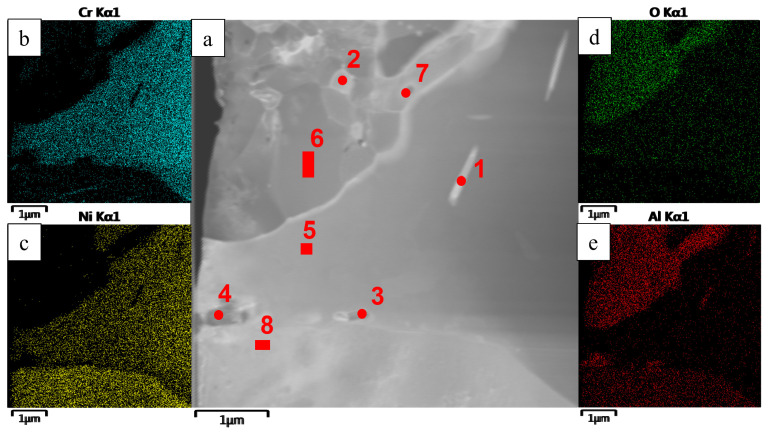
A TEM image (**a**) of a lamella cut from the oxidized layer of the *base+2.5%Mo-1.5%Ta-1.5%La-0.5%Ru* sample: and element distribution maps: (**b**) Cr; (**c**) Ni; (**d**) O; (**e**) Al.

**Figure 18 materials-16-03299-f018:**
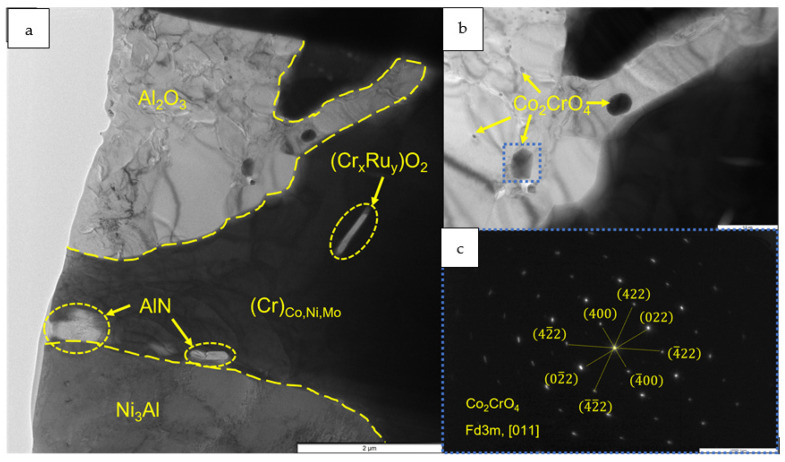
A TEM image of the structural components of a lamella cut from the oxidized layer of the *base+2.5%Mo-1.5%Ta-1.5%La-0.5%Ru* alloy: (**a**) the distribution of structural components in the lamella; (**b**) the zoomed-in area with inclusions of the Co_2_CrO_4_ phase; and (**c**) an electron diffraction pattern of Co_2_CrO_4_ recorded along the [011] zone axis.

**Figure 19 materials-16-03299-f019:**
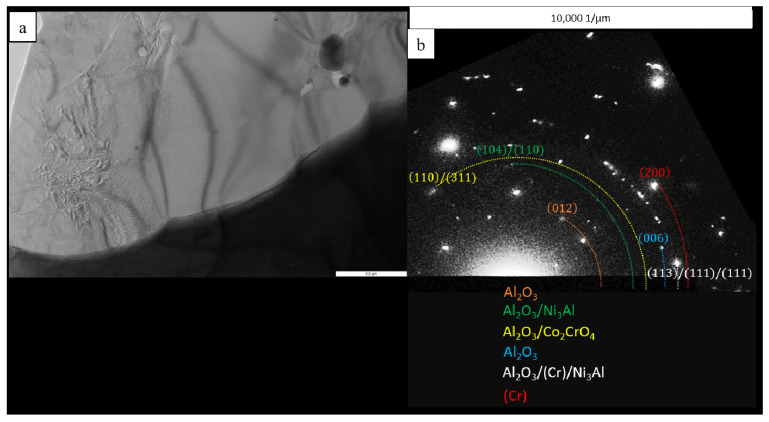
A TEM image of the structural components of the lamella cut from the oxidized layer of the *base+2.5%Mo-1.5%Ta-1.5%La-0.5%Ru* alloy: (**a**) the multiphase region; and (**b**) electron diffraction analysis in a polycrystalline lattice in this region.

**Figure 20 materials-16-03299-f020:**
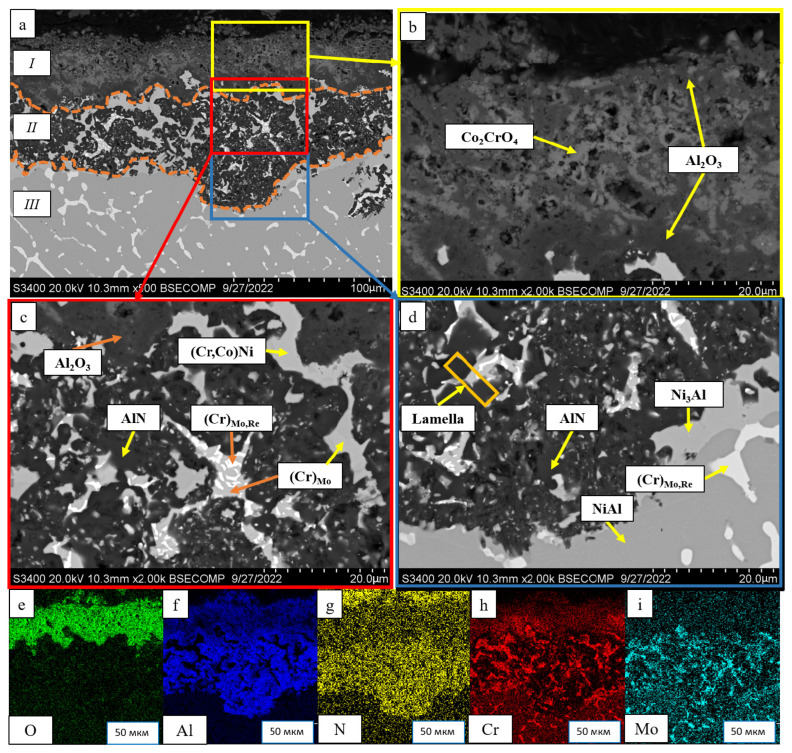
A SEM image of the surface of the oxidized sample with composition 4 (*base+2.5%Mo-1.5%Re-1.5%Ta-0.2%Ti*): (**a**) under magnification ×500; (**b**) the continuous Al_2_O_3_ oxide film with non-uniform distribution of Co_2_CrO_4_ spinel inclusions; (**c**) the transitional MeN-Me layer containing AlN inclusions; (**d**) the metal layer with sparse inclusions of AlN; the element distribution maps: (**e**) O; (**f**) Al; (**g**) N; (**h**) Cr; (**i**) Mo; and (**d**) the lamella cutting site.

**Figure 21 materials-16-03299-f021:**
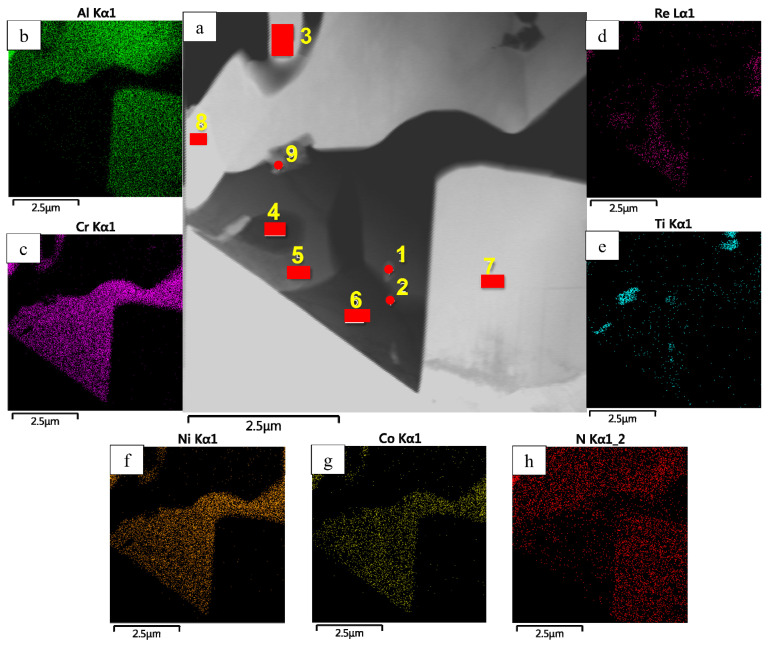
A TEM image (**a**) of the lamella cut from the oxidized layer of the *base+2.5%Mo-1.5%Re-1.5%Ta-0.2%Ti* sample of the MeN-Me transitional layer, and the maps of distribution of the major elements: (**b**) Al; (**c**) Cr; (**d**) Re; (**e**) Ti; (**f**) Ni, (**g**) Co; (**h**) N.

**Figure 22 materials-16-03299-f022:**
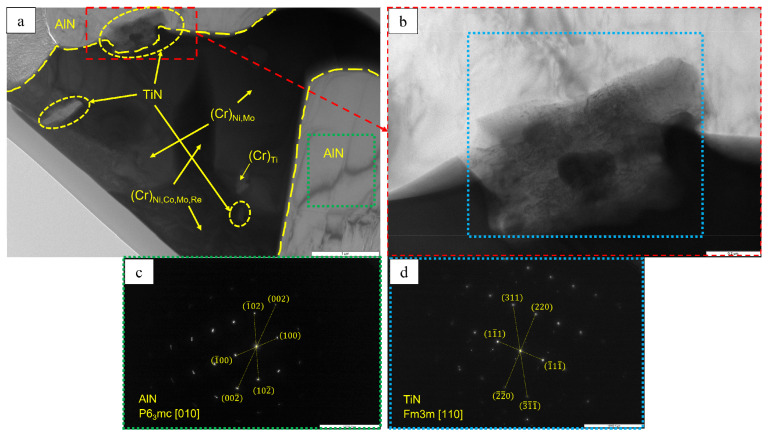
A TEM image of the structural components of the transitional layer of the *base+2.5%Mo-1.5%Re-1.5%Ta-0.2%Ti* sample: (**a**) distribution of the structural components in the lamella; (**b**) the zoomed-in area of the AlN phase; (**c**) an electron diffraction pattern recorded for the AlN grain along the [010] zone axis; and (**d**) an electron diffraction pattern recorded for the TiN grain along the [110] axis zone.

**Figure 23 materials-16-03299-f023:**
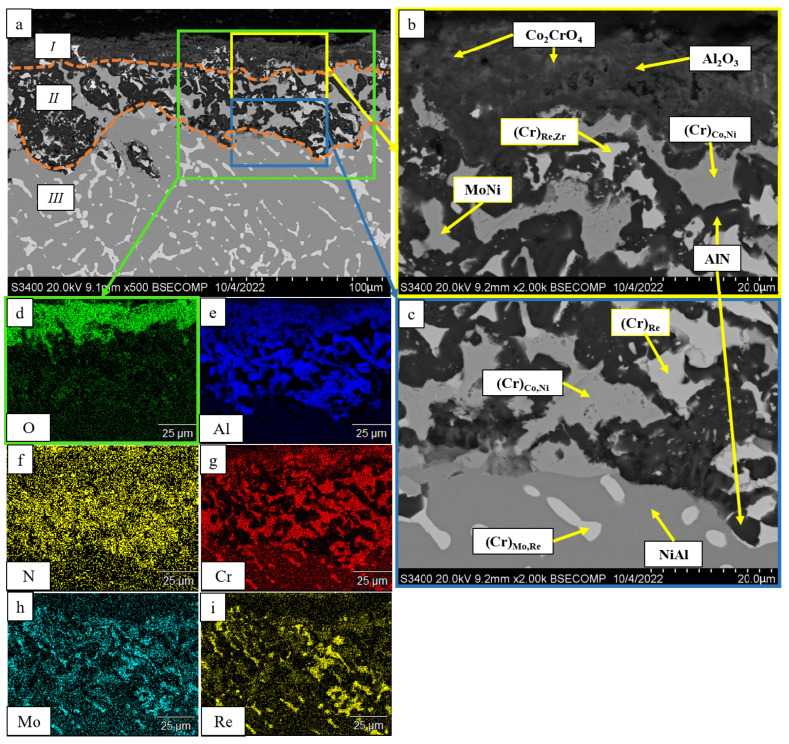
A SEM image of the surface of the oxidized sample with composition 5 (*base+2.5%Mo-1.5%Re-1.5%Ta-1.5%Zr*): (**a**) under magnification ×500; (**b**) the continuous Al_2_O_3_ oxide film with non-uniform distribution of Co_2_CrO_4_ spinel inclusions; (**c**) the transitional MeN-Me layer containing AlN inclusions; and the element distribution maps: (**d**) O; (**e**) Al; (**f**) N; (**g**) Cr, (**h**); Mo; (**i**) Re.

**Table 1 materials-16-03299-t001:** The calculated composition of the alloys.

No.	Composition of the Base	Composition of the Dopant (*X*), wt.%
1	NiAl-12Cr-6Co *(base)*	+2.5Mo-0.5Re-0.5Ta
2		+2.5Mo-1.5Re-1.5Ta
3		+2.5Mo-1.5Ta-1.5La-0.5Ru
4		+2.5Mo-1.5Re-1.5Ta-0.2Ti
5		+2.5Mo-1.5Re-1.5Ta-0.2Zr

**Table 2 materials-16-03299-t002:** Characteristics of the initial powders and modifying additives.

Compound	CAS No or Grade	Standard: GOST/TU/ISO/ASTM	Particle Size, µm	Chemical Composition, %
**Major components**				
NiO	1313-99-1	GOST 17607-72/ISO 12169	<40	99.00
Cr_2_O_3_	1308-38-9	TU 6-09-4272-84/ISO 4621	<20	99.00
Co_3_O_4_	1307-96-6	GOST 18671-73	<30	99.00
Al	P3	GOST 6058-73/ASTM B221-21 & B595-2	<140	98.00
Al	Al300	TU 48-5-226-87/ASTM B221-21	<50	99.70
**Modifying additives (MA)**				
MoO_3_	1313-27-5	TU 6-09-4471-77/ASTM A146-04	<50	99.00
Zr	702 (shaving)	TU 95.166-83/ASTM B551	≤600	99.80
Ta	EB	TU 48-19-72-92/ASTM B708-12	<20	98.00
Re	7440-15-5	TU 48-4-195-87/ASTM E696-07	<150	99.99
Ru	7440-18-8	GOST 12343-79/ASTM-B717	≤100	99.95
Ti	TF-0	TU 14-22-57-92/ASTM B367-00	≤30	99.80
La	F01	TU 1-92-200-2000	≤100	99.97

**Table 3 materials-16-03299-t003:** The impurity composition of the *base-X* alloys.

Element	Concentration, wt.%				
	**+2.5Mo-0.5Re-0.5Ta*	**+2.5Mo-1.5Re-1.5Ta*	**+2.5Mo-1.5Ta-1.5La-0.5Ru*	**+2.5Mo-1.5Re-1.5Ta-0.2Ti*	**+2.5Mo-1.5Re-1.5Ta-0.2Zr*
O	0.0261	0.0281	0.0136	0.0212	0.0321
N	0.0023	0.0029	0.0009	0.0013	0.0026
∑_gas impurities_	0.0283	0.0310	0.0145	0.0223	0.0346
∑_impurity elements **_	0.1449	0.1135	0.1395	0.1478	0.1613

* NiAl-12Cr-6Co (*base-X*); ** ∑_Mg,Na,Si,Ca,K,FeMn Cu,W,S,C_.

**Table 4 materials-16-03299-t004:** The phase composition of the *base+* alloy with modifying additives *X*.

No.	Modifying Additive X	Phase	Mass Fraction, %	Lattice Parameters, Å		
				a	b	c
1	2.5%Mo-0.5%Re-0.5%Ta	NiAl	100	2.867	-	-
2	2.5%Mo-1.5%Re-1.5%Ta	NiAl	80	2.879	-	-
		Ni(Al,X) *	14	2.931	-	-
		Ni(Al,Ta)	6	2.997	-	-
3	2.5%Mo-1.5%Ta-1.5%La-0.5%Ru	NiAl	74	2.882	-	-
		MoNi	18	9.089	9.084	8.834
		Ni(Al,Ta)	8	2.918	-	-
4	2.5%Mo-1.5%Re-1.5%Ta-0.2%Ti	NiAl	54	2.884	-	-
		Ni(Al,Ti)	46	2.927	-	-
5	2.5%Mo-1.5%Re-1.5%Ta-0.2%Zr	NiAl	86	2.896	-	-
		Ni_2_(Zr,Al)	14	2.983	-	-

* In addition to the major phase with the B2-type structure, there are phases with the same structure or bcc solutions; however, the phase composition cannot be determined by this method.

**Table 5 materials-16-03299-t005:** Mechanical properties of the *base+X* alloys.

No.	*base+X* Alloy	Hardness, HV	σ_ucs_, MPa	σ_ys_, MPa	*ε_pd_*, %
1	+2.5Mo-0.5%Re-0.5%Ta	5.15	1266	1117	<1
2	+2.5Mo-1.5%Re-1.5%Ta	4.94	924	-	<1 *
3	+2.5Mo-1.5%Ta-1.5%La-0.5%Ru	5.51	1241	-	<1 *
4	+2.5Mo-1.5%Re-1.5%Ta-0.2%Ti	5.56	1644	1518	<1
5	+2.5Mo-1.5%Re-1.5%Ta-0.2%Zr	5.12	1304	-	<1 *

* The samples have undergone brittle failure.

**Table 6 materials-16-03299-t006:** The effect of dopants on the oxidation kinetics of cast alloys.

No.	*base+X* alloy	Fitting Equation	Weight Gain, g/m^2^
1	+2.5Mo-0.5%Re-0.5%Ta	y = 0.0002x^3^ − 0.0076x^2^ + 0.231x + 2.2869	7.282
2.1	+2.5Mo-1.5%Re-1.5%Ta	y = 23.084ln(x) + 45.958	133.624
2.2	+2.5Mo-1.5%Re-1.5%Ta (TVT *)	y = − 0.0295x^2^ + 2.3405x + 3.6274	48.372
3	+2.5Mo-1.5%Ta-1.5%La-0.5%Ru	y = 0.0002x^3^ − 0.0094x^2^ + 0.165x + 1.096	2.772
4	+2.5Mo-1.5%Re-1.5%Ta-0.2%Ti	y = −0.044x^2^ + 3.1537x + 0.3412	55.886
5	+2.5Mo-1.5%Re-1.5%Ta-0.2%Zr	y = −0.0295x^2^ + 2.593x + 5.6503	57.546

*—TVT—thermo-vacuum treatment.

**Table 7 materials-16-03299-t007:** The phase composition of the oxidized layer of *base+X* alloys.

No.	*Base+X* Alloy	Phase	wt.%	Lattice Parameters, Å	
				a	c
1	+2.5%Mo-0.5%Re-0.5%Ta	Al_2_O_3_	84	4.772	13.044
		NiAl	11	2.878	-
		MoNi	4	3.644	-
		Ni_3_AlN	1	3.808	-
2	+2.5%Mo-1.5%Re-1.5%Ta	Al_2_O_3_	42	4.770	13.054
		Co_2_CrO_4_	40	8.168	-
		MoNi	13	3.598	-
		NiAl	5	2.887	-
3	+2.5%Mo-1.5%Ta-1.5%La-0.5%Ru	Al_2_O_3_	66	4.767	-
		Co_2_CrO_4_	13	8.135	-
		NiAl	13	2.868	-
		TaCo_2_	8	4.765	15.287
4	+2.5%Mo-1.5%Re-1.5%Ta-0.2%Ti	Al_2_O_3_	69	4.792	13.094
		Co_2_CrO_4_	20	8.163	-
		NiAl	6	2.884	-
		Ni_(1−x)_Al_(x)_	5	3.605	-
5	+2.5%Mo-1.5%Re-1.5%Ta-0.2%Zr	Al_2_O_3_	63	4.781	13.066
		Co_2_CrO_4_	30	8.163	-
		MoNi	7	3.605	-

**Table 8 materials-16-03299-t008:** The chemical composition (wt.%) of a lamella cut from the oxidized layer of the *base+2.5%Mo-1.5%Ta-1.5%La-0.5%Ru*.

Spectra	O	N	Al	Cr	Co	Ni	Mo	Ru
1	6.32	-	8.18	41.52	11.91	16.07	14.16	1.83
2	49.55	-	49.06	1.39	-	-	-	-
3	14.36	7.54	50.32	1.37	2.06	24.34	-	-
4	-	32.57	67.43	-	-	-	-	-
5	-	-	0.64	50.95	14.52	19.46	14.42	-
6	50.65	-	49.35	-	-	-	-	-
7	20.58	-	22.52	19.71	10.36	10.80	16.04	-
8	-	-	15.67	12.89	7.24	64.19	-	-

**Table 9 materials-16-03299-t009:** The chemical composition (wt.%) of a lamella cut from the oxidized layer of the *base+2.5%Mo-1.5%Re-1.5%Ta-0.2%Ti* sample of the MeN-Me transitional layer.

Spectra	N	Al	Ti	Cr	Co	Ni	Mo	Re
1	-	-	18.93	45.84	6.16	19.54	9.55	-
2	1.91	0.34	4.42	39.57	9.56	16.15	13.57	14.48
3	24.58	78.06	-	0.36	-	-	-	-
4	-	-	-	37.41	10.60	17.47	15.16	19.36
5	1.87	-	-	51.61	7.96	25.89	12.68	-
6	-	-	-	35.57	10.44	17.34	14.69	21.97
7	23.39	76.61	-	-	-	-	-	-
8	23.16	79.84	-	-	-	-	-	-
9	24.37	-	63.48	9.64	0.73	1.80	-	

## Data Availability

Data is contained within the article.

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
