# Peer review of "The Effect of Dopants on Structure Formation and Properties of Cast SHS Alloys Based on Nickel Monoaluminide"

_materials, 2023, doi:10.3390/ma16093299_

Round 1

Reviewer 1 Report

Some comments and suggestions to authors:

1. Most figures in this paper are composed of several figures. Each figure should be marked in *a), (b), (c)...etc., and added figure captions for them.

2. The composition in Figs.16 and 20 should be a new Table in the revised manuscript.

3. In Table 7, I suggest using "wt.%" to replace "Wt.%".

4. The full name of TEM should be added in the Abstract.

5. The expression of the alloy of NiAlCrCo or AlAl-Ct-Co should be consistent.

6. What is the base alloy in Table 1, including alloys 2 to 5?

7. What is X in Eq. 1?

8. Detailed information on the instrument used in this study should be described.

9 The line in each figure is not clear.

Author Response

We would like to express our thanks to the reviewer for a careful study of our paper and useful recommendations. Obviously, they will improve the quality of our paper. Please find below our point-by-point responses to the comments. All changes in manuscript text were embedded and highlighted in yellow. Revised text contains corrections, made upon reviewer’s recommendations.

  1. Most figures in this paper are composed of several figures. Each figure should be marked in *a), (b), (c)...etc., and added figure captions for them.

This remark was considered; the notation and description of the figures were added.

  1. The composition in Figs.16 and 20 should be a new Table in the revised manuscript.

New tables have been added in the revised version of the manuscript.

  1. In Table 7, I suggest using "wt.%" to replace "Wt.%".

This remark was taken into account and the correction was made to the text of the article

  1. The full name of TEM should be added in the Abstract.

This remark was taken into account and the correction was made to the text of the article

  1. The expression of the alloy of NiAlCrCo or AlAl-Ct-Co should be consistent.

This remark was taken into account and the correction was made to the text of the article

  1. What is the base alloy in Table 1, including alloys 2 to 5?

The basic part of the alloy (NiAl-12Cr-6Co) is presented – the "base" for all alloys.

  1. What is X in Eq. 1?

X is composition of the dopant (Mo, Re, Ta, La, Ru, Ti, Zr).

  1. Detailed information on the instrument used in this study should be described.

Sample preparation, powder characteristics and the scheme of the SHS process were added.

9 The line in each figure is not clear.

This remark was taken into account and the correction was made to the text of the article

Reviewer 2 Report

This manuscript presents an investigation of Alloys based on NiAlCrCo (base) with complex dopants (base+2.5Mo-0.5Re-0.5Ta, base+2.5Mo-1.5Re-1.5Ta, base+2.5Mo-1.5Ta-1.5La-0.5Ru, base+2.5Mo-1.5Re-1.5Ta-0.2Ti, base+2.5Mo-1.5Re-1.5Ta-0.2Zr), which were fabricated by centrifugal high-temperature-synthesis metallurgy. The authors studied the phase and impurity compositions, structure, mechanical properties, and the mechanism of high-temperature oxidation at T =1150°C. The topic is interesting due to it has great interest to produce heat-resistant alloys for the components of gas-turbine engines. The results found are consistent and makes a good impression, which is known for all least one of the authors.

The paper could be published. However, there are a few points that should be attended, which I feel, they would improve the presentation.

-          Page 1, abstract, line 6 from the top. Remove “found to be”.

-          Page 1, abstract, line 9 from the top. “proved” should be “observed”.

-          Page 1, abstract, line 13 from the top. Remove “respectively”.

-          Page 6, line 3 from the top. “demonstrates” should be “shows or displays”.

-          Page 9, line 2 from the bottom. “demonstrates” should be “showed”.

-          Page 16, line 9 from the bottom. “demonstrates” should be “showed”.

-          Page 24, line 3 from the bottom. “detected” should be “observed”.

-          Page 26, line 4 from the top. “demonstrated” should be “observed”.

Author Response

We would like to express our thanks to the reviewer for a careful study of our paper and useful recommendations. Obviously, they will improve the quality of our paper. Please find below our point-by-point responses to the comments. All changes in manuscript text were embedded and highlighted in yellow. Revised text contains corrections, made upon reviewer’s recommendations.

-          Page 1, abstract, line 6 from the top. Remove “found to be”.

-          Page 1, abstract, line 9 from the top. “proved” should be “observed”.

-          Page 1, abstract, line 13 from the top. Remove “respectively”.

-          Page 6, line 3 from the top. “demonstrates” should be “shows or displays”.

-          Page 9, line 2 from the bottom. “demonstrates” should be “showed”.

-          Page 16, line 9 from the bottom. “demonstrates” should be “showed”.

-          Page 24, line 3 from the bottom. “detected” should be “observed”.

-          Page 26, line 4 from the top. “demonstrated” should be “observed”.

All recommendations have been taken into account and all the points have been corrected.

Reviewer 3 Report

In this work, the authors studied the microstructure and oxidation behavior of Ni-Al-Cr-Co alloys alloyed with Mo, Re, Ta, Ru, Ti,, Ti and Zr. The alloys were prepared by self-propagating high temperature synthesis (SHS), namely, by reacting metallic oxides with Al (aluminothermic reaction). The paper is relatively complex, as it studies five different alloy compositions. It provides a substantial number of new results. However, it lacks a thorough discussion and comprehension of the results. It needs to be major-revised.

1.The authors should provide a schematic of the SHS process. It is not clear how ingots were produced from the starting powders.

2.Al2O3 is one of the products of SHS process (Equation 1). You claim that it formed a protective scale on the alloys (Fig. 12,14, 15, etc.). However, some Al2O3 might already have been there from the synthesis step alone. Have you removed Al2O3 from the prepared alloys prior to oxidation or not?

3.The denomination of the purity grades (second column in Table 2) needs an explanation. What is “PA-4”, “ASD-1”, “E635”, etc. You should convert these symbols into weight percentages and clearly state the impurities present in the starting powders.

4.It is not clear why two different Al powders were used during the synthesis (Table 2). You need to explain it.

5.It is recommended to simplify the denomination of the alloys since 2.5Mo and 1.5Ta were present in most of the alloys (Table 1). It will help the readers to see the differences between the alloys clearly.

6.Powder diffraction file numbers of identified phases should be included in Figs. 1 and 10.

7.You should include EDS maps for Al in Figs. 3-6 since you claim that NiAl was present in the alloys.

8.Several kinetic curves in Fig. 8 indicate a non-zero intercept at t=0 h. What was the heating rate used during the oxidation annealing?

9.The alloying elements have a profound effect on the overall mass gain (Fig. 8). The difference between the alloys is nearly 2 orders of magnitude. It could be related to the structure of the Al2O3 layer. You should estimate parabolic rate constants from the data and compare them with previously studied alumina-forming alloys. It is possible that Al2O3 was not fully covering some of the alloys.

10.You should provide a schematic of the oxidation mechanism and explain the effects of various elements (La, Ru, Ti, Zr) on the protective scale formation.

11.The paper has a relatively high number of self-citations (13 papers out of 30, i.e., 43%). You should also compare your results with results from different groups working in the field of superalloys.

Author Response

We would like to express our thanks to the reviewer for a careful study of our paper and useful recommendations. Obviously, they will improve the quality of our paper. Please find below our point-by-point responses to the comments. All changes in manuscript text were embedded and highlighted in yellow. Revised text contains corrections, made upon reviewer’s recommendations.

1.The authors should provide a schematic of the SHS process. It is not clear how ingots were produced from the starting powders.

This remark was considered, the scheme of the SHS process was added (Figure 1)

2.Al2O3 is one of the products of SHS process (Equation 1). You claim that it formed a protective scale on the alloys (Fig. 12,14, 15, etc.). However, some Al2O3 might already have been there from the synthesis step alone. Have you removed Al2O3 from the prepared alloys prior to oxidation or not?

Complete phase separation was achieved due to effect of centrifugal forces on SHS casting at optimal conditions. Therefore, there were no particles of aluminum oxide in the cast ingot of the target products. Al2O3 layer on the surface of the intermetallic sample appeared result in the oxidation.

3.The denomination of the purity grades (second column in Table 2) needs an explanation. What is “PA-4”, “ASD-1”, “E635”, etc. You should convert these symbols into weight percentages and clearly state the impurities present in the starting powders.

International standards for the grades were added in Table 2. The purity is presented in column 5.

4.It is not clear why two different Al powders were used during the synthesis (Table 2). You need to explain it.

Different grades of aluminum are used to control SHS processes. The information has been added to the text article with a link to the source.

5.It is recommended to simplify the denomination of the alloys since 2.5Mo and 1.5Ta were present in most of the alloys (Table 1). It will help the readers to see the differences between the alloys clearly.

The base part of the alloy (NiAl12Cr6Co) is presented – «base». Influence of other additives was studied and is written in the text and figures.

  1. Powder diffraction file numbers of identified phases should be included in Figs. 1 and 10.

The numbers of identified phases have been added.

7.You should include EDS maps for Al in Figs. 3-6 since you claim that NiAl was present in the alloys.

EDS maps for Al have been added

8.Several kinetic curves in Fig. 8 indicate a non-zero intercept at t=0 h. What was the heating rate used during the oxidation annealing?

In order to study the effect of complex doping on high-temperature oxidation resistance of the alloys, air annealing at 1150°C was performed during 30 h; the samples were periodically weighed. The measurements were carried out on samples of the appropriate exposure to temperature. The first measurement was carried out after 2 hours of holding at a temperature of 1150 . The heating rate to the desired temperature was V = 12 /min. The zero point of the experiment is reaching a T= 1050 .

9.The alloying elements have a profound effect on the overall mass gain (Fig. 8). The difference between the alloys is nearly 2 orders of magnitude. It could be related to the structure of the Al2O3 layer. You should estimate parabolic rate constants from the data and compare them with previously studied alumina-forming alloys. It is possible that Al2O3 was not fully covering some of the alloys.

When describing each structure of the oxidized layer, the expected character of oxidation is indicated. In the first hours, the sample is covered with a layer and, on some compositions, the Al2O3 layer is destroyed from the inside, for a while, access to new active oxygen and nitrogen ion is opened. This is how stepwise oxidation occurs. This information is described in detail in the article.

10.You should provide a schematic of the oxidation mechanism and explain the effects of various elements (La, Ru, Ti, Zr) on the protective scale formation.

The oxidation scheme is added to Figure 12 with a schematic description of the influence of each element on the oxidation process. Details for each alloy are described in the text of the SEM study.

11.The paper has a relatively high number of self-citations (13 papers out of 30, i.e., 43%). You should also compare your results with results from different groups working in the field of superalloys.

The remark was taken into account and the list of reference was expanded.

Round 2

Reviewer 1 Report

The revised manuscript is suitable to be published in Materials.

Author Response

We would like to express our thanks to the reviewewr for thorough study of our manuscript.

Reviewer 3 Report

The authors answered most of my comments. The paper has been improved. It is publishable subject to minor revision.

1.I couldn’t find the diffraction file numbers of the phases in the revised paper. Include them in figure captions (Fig. 2, 11).

2.Alloys 1, 2.2, 3, 4, and 5 followed a parabolic oxidation kinetics (Fig. 9, Table 6, line 265). Therefore, parabolic rate constants should be estimated. The kp values should be compared with previously studied alloys, see, e.g. https://doi.org/10.1007/s11661-022-06860-6, https://doi.org/10.1038/s41529-021-00184-3, etc.

Author Response

We would like to express our thanks to the reviewer for a careful study of our paper and the comments. Please find below our point-by-point responses to the comments. All changes in manuscript text were embedded. Revised text contains corrections, made upon reviewer’s recommendations.

  1. I couldn’t find the diffraction file numbers of the phases in the revised paper. Include them in figure captions (Fig. 2, 11).

Diffraction file numbers of the phases were added in Figures 2 and 11.

  1. Alloys 1, 2.2, 3, 4, and 5 followed a parabolic oxidation kinetics (Fig. 9, Table 6, line 265). Therefore, parabolic rate constants should be estimated. The kp values should be compared with previously studied alloys, see, e.g. https://doi.org/10.1007/s11661-022-06860-6, https://doi.org/10.1038/s41529-021-00184-3, etc.

The oxidation constant values were calculated and added as a graph in Figure 9. The links you provided to the work are of interest to the study. Information about the relevance of these works with links to these articles has been added to the introduction. Intermetallic alloys were tested at 1150 ℃ of oxidation, but superalloys - at 1000 ℃.
